# Factors Influencing Pharmacokinetics of Tamoxifen in Breast Cancer Patients: A Systematic Review of Population Pharmacokinetic Models

**DOI:** 10.3390/biology12010051

**Published:** 2022-12-28

**Authors:** Jaya Shree Dilli Batcha, Arun Prasath Raju, Saikumar Matcha, Elstin Anbu Raj S., Karthik S. Udupa, Vikram Gota, Surulivelrajan Mallayasamy

**Affiliations:** 1Department of Pharmacy Practice, Manipal College of Pharmaceutical Sciences, Manipal Academy of Higher Education, Manipal 576 104, Karnataka, India; 2Public Health Evidence South Asia, Department of Health Information, Prasanna School of Public Health, Manipal Academy of Higher Education, Manipal 576 104, Karnataka, India; 3Department of Medical Oncology, Kasturba Medical College, Manipal Academy of Higher Education, Manipal 576 104, Karnataka, India; 4Department of Clinical Pharmacology, ACTREC, Tata Memorial Centre, Mumbai 410 210, Maharashtra, India; 5Center for Pharmacometrics, Manipal Academy of Higher Education, Manipal 576 104, Karnataka, India

**Keywords:** tamoxifen, systematic review, population pharmacokinetics, breast cancer

## Abstract

**Simple Summary:**

Breast cancer is the most common type of cancer in women. Tamoxifen is the most preferred drug used to treat breast cancer. It has been reported that tamoxifen and its metabolites have significant variability in their pharmacokinetics. This systematic review identified five population pharmacokinetic model studies for tamoxifen. These studies were summarized, and various factors affecting tamoxifen’s and its metabolites pharmacokinetics have been reported in this review. Most studies reported a two-compartment model with first-order absorption and elimination. Various factors, such as genetic variation, age, gender, BMI, co-medication, and postmenopausal status are reported to affect the disposition of tamoxifen and its metabolites. So, while addressing the pharmacokinetic variability of this drug, all these factors must be considered. These models should be externally evaluated to verify the model’s generalizability and for model-informed dosing in the clinical setup.

**Abstract:**

Background: Tamoxifen is useful in managing breast cancer and it is reported to have significant variability in its pharmacokinetics. This review aimed to summarize reported population pharmacokinetics studies of tamoxifen and to identify the factors affecting the pharmacokinetics of tamoxifen in adult breast cancer patients. Method: A systematic search was undertaken in Scopus, Web of Science, and PubMed for papers published in the English language from inception to 20 August 2022. Studies were included in the review if the population pharmacokinetic modeling was based on non-linear mixed-effects modeling with a parametric approach for tamoxifen in breast cancer patients. Results: After initial selection, 671 records were taken for screening. A total of five studies were selected from Scopus, Web of Science, PubMed, and by manual searching. The majority of the studies were two-compartment models with first-order absorption and elimination to describe tamoxifen and its metabolites’ disposition. The CYP2D6 phenotype and CYP3A4 genotype were the main covariates that affected the metabolism of tamoxifen and its metabolites. Other factors influencing the drug’s pharmacokinetics included age, co-medication, BMI, medication adherence, CYP2B6, and CYP2C19 genotype. Conclusion: The disposition of tamoxifen and its metabolites varies primarily due to the CYP2D6 phenotype and CYP3A4 genotype. However, other factors, such as anthropometric characteristics and menopausal status, should also be addressed when accounting for this variability. All these studies should be externally evaluated to assess their applicability in different populations and to use model-informed dosing in the clinical setting.

## 1. Introduction

Cancer is broadly described as unregulated cell proliferation that invades other tissues and leads to death if left untreated. Cancer is classified into more than 100 types and named based on the organ or origin of cancer formation [1]. According to IARC (International Agency for Research on Cancer)’s estimation in 2018, there were 9.5 million deaths due to cancer and 17.0 million new cancer cases worldwide. By 2040, the global burden is expected to grow to 16.3 million deaths and 27.5 million new cancer cases because of the growth and aging of the population [2].

Breast cancer develops in the breast or mammary gland [3]. Breast cancer is the most frequently diagnosed cancer in women, accounting for 24.5% of all cancer cases globally. According to the American Cancer Society, there is a 1.7% increase in the incidence rate of breast cancer in the Asian/Pacific Islander population, followed by 0.4% and 3% increases in the non-Hispanic black and Hispanic populations [4]. These statistics demonstrate the need for rigorous research to prevent and treat breast cancer.

The treatment options for the management of breast cancer consists of radical mastectomy, chemotherapy, targeted molecular/endocrine therapy, and radiation therapy. Radical mastectomy followed by radiation and chemotherapy is the standard therapy for breast cancer. Adjuvant endocrine therapy has been reported to reduce the recurrence and improve the overall survival rate in women. [5,6].

Tamoxifen is a selective estrogen receptor modulator that acts by inhibiting the proliferative activities of estrogen on the mammary epithelium in breast tissue [7]. Tamoxifen is a prodrug actively utilized in treating estrogen receptor (ER)-positive breast cancer [8]. It is converted to the two primary active metabolites, endoxifen (4-hydroxy-N-desmethyl tamoxifen) and 4-hydroxy-tamoxifen, through CYP2D6 and CYP3A4/5 enzymes. These tamoxifen metabolites (endoxifen and 4-hydroxy tamoxifen) have a 100-fold higher affinity for ER and a 30–40-fold increase in antiestrogenic activity [9].

Tamoxifen is usually administered as tamoxifen citrate, whereas endoxifen is taken orally as Z-endoxifen hydrochloride. Tamoxifen typically has an elimination half-life of 5–7 days. The half-life of endoxifen ranges from 49.0 to 68.1 h for doses ranging from 20 and 160 mg. Tamoxifen has a prolonged Tmax of 4–7 h, and endoxifen has a shorter Tmax of 2–4 h. Tamoxifen and endoxifen have markedly different Cmax values. The Cmax for a 20 mg single dosage of tamoxifen is 40 ng/mL, but the Cmax values for 20 and 160 mg single doses of Z-endoxifen hydrochloride are 64.8 and 635 ng/mL, respectively [10,11].

The pharmacokinetics (PK) of tamoxifen and its active metabolite, endoxifen, have been found to exhibit high interindividual variability (IIV). This variability has been influenced in part by the genetic variation in the metabolic enzymes CYP2D6, CYP3A4/5, CYP2B6, CYP2C19, and other non-genetic factors [12,13,14,15,16]. Endoxifen concentration should be higher than 5.97 ng/mL to produce the desired clinical effect, and in those with concentrations below this threshold, there is high probability of recurrence [17]. Tamoxifen has many benefits, but 30–40% of patients experience therapy failure because of tamoxifen resistance [18]. Tamoxifen resistance is developed by various mechanisms which include changes in the function and the structure of estrogen receptor, metabolizing pathway, and the tumor environment [19].

CYP2D6 plays a vital role in the conversion of tamoxifen to its active metabolite, endoxifen. When taking into account the various ethnic groups within a population, this enzyme is highly polymorphic [20,21]. The CYP2D6 gene, according to the literature, has about 100 allelic variants that result in various phenotypic patterns. The population is divided into four groups based on these phenotypes: poor metabolizers (PMs), intermediate metabolizers (IMs), normal metabolizers (NMs), and ultrarapid metabolizers (UMs). It has been noted in several pharmacokinetic studies that a population with decreased CYP2D6 enzyme activity has a negative impact on the plasma concentration of endoxifen [22,23,24].

Population pharmacokinetics (PopPK) intends to characterize the observed interindividual variability (IIV) in drug exposure for a specific population sample. The method estimates the population mean (θ) and IIV (η) of pharmacokinetics (PK) parameters, as well as the remaining unexplained, or residual, variability (ε). Covariates such as genetic polymorphism, demographics, and other pathophysiological variables identified using the PopPK approach can help in addressing the variability in the PK of drugs [25].

This study aims to summarize and explore the variables affecting the pharmacokinetics of tamoxifen from previously published PopPK articles of tamoxifen in breast cancer patients.

## 2. Methods

### 2.1. Literature Search

The Preferred Reporting Items for Systematic Reviews and Meta-analyses (PRISMA) criteria [26] were utilized to conduct the systematic search in the PubMed, Scopus, and Web of Science databases. The review was performed in accordance with the PRISMA guidelines. The review comprised articles published from inception until 20 August 2022. “Tamoxifen” AND “Pharmacokinetics” OR “Population Pharmacokinetics” OR “PopPK” OR “Nonlinear mixed effects modeling” OR “NONMEM” AND “Breast Neoplasms” OR “Breast Cancer” were the search keywords employed in the databases. Appendix A depicts the search strategy used in the various databases.

### 2.2. Study Selection Criteria

A systematic search was undertaken independently by two reviewers (J.S.D.B. and A.P.R.) to identify relevant literature. The reviewers assessed the titles, abstracts, and full texts of the publications using the predefined inclusion and exclusion criteria. If a conflict was noted during the search and review, a third reviewer was consulted to resolve the issue to reach an agreement. The following criteria were considered to select studies: (i) the parametric approach, (ii) the non-linear mixed-effects modeling approach used in PopPK analysis, and (iii) a tamoxifen model developed in breast cancer patients. The studies were excluded if (i) the study population was other than breast cancer patients, and (ii) conference abstracts, editorials, letters, reviews, short communications, books, and book chapters were omitted.

## 3. Data Extraction

Two authors (J.S.D.B. and A.P.R.) independently extracted data from the full texts of the selected articles using a standardized extraction form and then cross-checked it. M.S.R. rectified data discrepancies. For each of the articles that were chosen, the following information was extracted: the first author, year of publication, sample size of the study, type and number of samples utilized for tamoxifen modeling, body weight, age, ethnicity, CYP2D6 phenotype, CYP3A4 genotype, menopausal status, PopPK software, bio-analytical method, PopPK estimates, structural model, residual variability, external validation, CYP2D6 phenotype-based metabolism, and covariates that have major impacts on the PopPK model. M.S.R. evaluated the extracted data. Microsoft Word was used to produce a PRISMA flowchart depiction.

### Data Quality Assessment

An adopted checklist established from recently reported guidelines for (i) population pharmacokinetic–pharmacodynamic studies [27], (ii) clinical pharmacokinetics [28], and (iii) three studies that integrated the (i) and (ii) checklists were used to evaluate the reporting quality of the selected PopPK studies for tamoxifen [29,30,31]. The integrated revised checklist had 37 criteria, which were divided into five categories, as shown in Table 1: title, abstract, introduction/background, results/methods, and conclusion/discussion. If the study’s relevant data could be identified, each criterion was assigned a score of 1; otherwise, it was given a score of 0. All the PopPK studies that were chosen were evaluated using these criteria. Each study’s compliance rate was determined using the formula below, and the result was shown as a percentage.
(1)$$\mathrm{Compliance}\;\mathrm{rate}\;(\%)=\frac{Total\;number\;of\;criteria\;satisfied}{Total\;number\;of\;criteria}*100$$

## 4. Results

### 4.1. Literature Search

Out of 2011 articles identified, 736, 551, and 724 were from Scopus, Web of Science, and PubMed databases, respectively. An additional article was identified through the manual search process. A total of 1035 duplicate articles were removed. After title and abstract screening, 654 articles were removed, and 17 were available as full-text articles. Out of 17 articles, 12 were excluded for the following reasons: (i) a linear regression model for pharmacogenetic analysis (*n* = 4); (ii) a linear mixed effect model (*n* = 1); (iii) a mixed normal model (*n* = 1); (iv) a TDM study (*n* = 1); (v) the patient population of the study was not with breast cancer (*n* = 1); (vi) meta-analysis (*n* = 1); (vii) only simulation was carried out (*n* = 3). There were five articles that remained for the systematic review. Figure 1 presents the PRISMA flow diagram describing the selection of the PopPK tamoxifen studies for the systematic review.

### 4.2. Quality Evaluation of Selected Literature

According to the PopPK study’s standards, three studies had a compliance rate above 80%, but two studies had less than 80% compliance (interval: 75.6–83.3%). Co-administration of the medication (20%), statistical method and software used (20%), equations for model structure and covariate relationships (40%), and study limitations (40%) were the criteria with the lowest compliance, as shown in Table 1. None of the studies had reported concentration vs. time plots in their articles.

### 4.3. Population Studied and Sample Size

Of the five studies, Schoell et al. and Schulze et al. were carried out by pooling data from various clinical trials conducted on diverse populations. White people, Africans, Middle Eastern Arabs, Asians, and Indians represented the population of those studies. Ter Heine et al.’s study was conducted in the Dutch population, and Puszkiel et al.’s in the European population. Dahmane et al.’s study population contained Caucasians, North Africans, and Indians. The sample size of the population ranged from 40 to 928, as shown in Table 2.

### 4.4. Sampling Procedure

Some studies collected a small number of samples from breast cancer patients, whilst others collected five or more samples from a patient on a single day and/or on different days. Blood was drawn at random and/or predetermined time intervals during the sampling. In a PopPK study by ter Heine et al., nine samples were taken at pre-dose, 30 min, 1, 1.5, 2, 4, 8, and 24 h after tamoxifen administration. Dahmane et al. collected sparse sampling on five different occasions. Blood samples were taken by Puszkiel et al. at pre-dose and subsequently every six months for three years. Schoell et al. had dense and sparse sampling ranging from a minimum of one to nine samples in a single subject. The number of samples per study and patients varied from 680 to 27,433 and 1 to 9, as shown in Table 2.

### 4.5. CYP2D6 SNPs and Genotype

All the studies analyzed the single-nucleotide polymorphisms (SNPs) in CYP2D6 gene and were classified into various categories of metabolizers based on functional alleles, diplotype, and antiestrogen activity score (AAS). The various categories of metabolizers are ultrarapid metabolizers, intermediate metabolizers, normal metabolizers, and poor metabolizers. Dahmane et al. [35] and ter Heine et al. [15] used a dextromethorphan-derived approach to determine the metabolic phenotype of CYP2D6 and CYP3A4/5 activity for the study population. In Puszkiel et al. [34], individuals were genotyped based on the presence of SNPs in CYP2D6 allele: *XN, *4, *5, *6, *7, *9, *10, *17, *41. So, based on these CYP2D6 alleles, the individual was assigned with a phenotype and score according to its activity, which was proposed by CPIC guidelines [36]. Schoell et al. [32] and Schulze et al. [33] assigned CYP2D6 phenotypes based on the genetic polymorphisms (SNPs) or genotype information available for each patient.

### 4.6. Bioanalytical Methods

Ter Heine et al. [15] and Puszkiel et al. [34] measured the plasma concentration of tamoxifen and its metabolites using ultrarapid-high-performance liquid chromatography–tandem mass spectrometry (UPLC-MS/MS). Dahmane et al. [35] analyzed the samples using the high-performance liquid chromatography–tandem mass spectrometry (HPLC-MS/MS) method. However, Schulze et al. [33] and Schoell et al. [32] employed UPLC-MS/MS and HPLC-MS/MS, respectively.

### 4.7. Population Pharmacokinetic Modeling

Most studies (*n* = 3) used NONMEM for PopPK modeling [15,34,35]. Three PopPK studies used a two-compartment model to explain the structural model of tamoxifen and its metabolites, while the other two used four- and seven-compartment models, respectively. In all the studies, tamoxifen and its metabolites’ disposition were described by first-order absorption and elimination accurately. The linear conversion of tamoxifen to its other metabolites was used in every study. Ter Heine et al., Schoell et al., and Schulze et al. used absorption lag time (t_lag_) to explain the delay in the absorption of the drug, and it varied from 0.389 to 0.455 h. The absorption rate (K_a_) ranged from 0.7 to 1.9 h^−1^, as shown in Table 3 [15,32,33,34,35]. Tamoxifen’s volume of distribution (Vd) during the steady state in the central compartment for the two-compartment model ranged from 753 to 1120 L, whereas it was 724 L in the four-compartment model and 1380 L in the seven-compartment model. The Vd and CL for endoxifen were fixed as 400 L and 5.1 L h^−1^ in two studies (Schoell et al. and Schulze et al.), taken from a clinical study that only administered endoxifen [37]. The proportional error model explained the residual variability in each of the five studies.

### 4.8. Influence of CYP2D6 Phenotype on Tamoxifen Metabolism

According to two studies, the CYP2D6 phenotype impacted the metabolic rate constant for the conversion of tamoxifen into 4-hydroxy tamoxifen (4-OHTam) and N-desmethyl tamoxifen (NDTam) into endoxifen [34,35]. Ter Heine et al. found that CYP2D6 metabolic phenotypes accounted for 54% of the variation in endoxifen formation. In two-compartment models, the CYP2D6 phenotype was identified as a covariate in endoxifen formation (CL23/F). The CYP2D6 phenotype explained 17% of IIV on the rate constant of endoxifen formation [35].

### 4.9. Influence of Other Covariates on PK Parameters of Tamoxifen and Its Metabolites

Tamoxifen’s PK parameters are influenced by a number of covariates, including CYP3A4 activity, CYP2C19, CYP2B6 polymorphism, medication adherence, age, co-medication, and body weight. Studies have reported that age and body weight affect the clearance and metabolism of tamoxifen. In the age range of 32 to 78, the relative tamoxifen clearance decreased by 9% for every increase in ten years of age. Non-adherence to medication has led to poor treatment outcomes for the patients [35]. Puszkiel et al. [34] reported that the population with CYP3A4*22 carriers had 23% and 19% decreases in the conversion rate constant from tamoxifen to N-desmethyl tamoxifen (K_TAM/NDT_) and elimination rate constant of N-desmethyl tamoxifen (K_e,NDT_). Patients with CYP2C19*2 alleles showed a 13% reduction in conversion rate constant from tamoxifen to 4-hydroxy tamoxifen (K_TAM/4-OHTAM_), and the presence of the CYP2B*6/*6 genotype was associated with a 23% decrease in the conversion rate constant from tamoxifen to tamoxifen-N-oxide (K_TAM/NOX-TAM_). A study reported that CYP3A4, age, and medication adherence explained 4%, 8%, and 1% of IIV in tamoxifen apparent clearance (CL_TAM_/F), and it increased linearly with increasing CYP3A4 activity and decreased linearly with age. This signifies that the reported model was able to fit well for their given data with IIV less than 20%. CYP2D6 enzyme inhibitors and CYP3A4 enzyme activity explained 6% and 4% of IIV on metabolic rate constant from tamoxifen to N-desmethyl tamoxifen [35]. CYP2D6 inhibitors were responsible for the reduced endoxifen formation rate with 87%, while the non-CYP2D6 inhibitors only accounted for 45%. CYP3A4/5 genotypes accounted for approximately 92% of tamoxifen metabolism [35].

### 4.10. External Validation

External validation was performed only in the study by Schoell et al. In this study, the final model was used to predict the concentrations of tamoxifen and endoxifen for an evaluation dataset with 936 subjects and was compared against the observed concentrations. The precision and bias of this PopPK model were evaluated using the mean prediction error (MPE) and the mean absolute prediction error (MAPE). The MPE for the model represented a low bias of −13.9 ng/mL for tamoxifen and a minimal bias of −0.923 ng/mL for endoxifen. The precision of the model was evaluated using MAPE, which was in the acceptable range of <8%, i.e., 7.62% for tamoxifen and 6.29% for endoxifen [32].

### 4.11. Simulation

Simulations were performed in three studies from the final model and the estimated parameter. Schoell et al., in the simulation with the CYP2D6 phenotypes as a covariate, discovered that 36% of PMs, 4.6% of IMs, and 0.60% of NMs had sub-target endoxifen steady-state concentrations [32]. In Schulze et al. [33] and Puszkiel et al. [34], a large virtual population was developed, which represented the patient characteristics of their study population, including the variability in the covariates. Using these developed data, model, and their estimated parameters, the simulations were performed. The simulation reported that the “one-dose-fits-all” strategy does not fit for all patients, and the tamoxifen dosing should be based on the CYP2D6 AAS and various other non-genetic factors to attain endoxifen target concentration.

## 5. Discussion

To the best of our knowledge, this is the first review summarizing data concerning the previously published population pharmacokinetics model for tamoxifen in breast cancer. There are many studies stating that categories of CYP2D6 phenotypes affect the transformation of the tamoxifen to its active metabolite endoxifen [38,39,40]. Based on the different CYP2D6 phenotype categories, each group is given a score for its antiestrogenic activity. For each class of CYP2D6 phenotypes, the allelic combination and AAS are as follows: in UMs, completely active alleles are duplicated (AAS: >2.0); NMs have two fully active alleles (AAS: 1.5–2.0); IMs have a combination of one low-activity allele or two low-activity alleles and one inactive allele (AAS: 0.5); PMs have two non-functional alleles (AAS: 0); and heterozygous extensive metabolizer (hetEM), a fifth phenotype, also consists of one fully active allele and one inactive allele [41].

However, recent large prospective clinical trials, i.e., Breast International Group (BIG) 1–98 and Arimidex, Tamoxifen, Alone or in Combination (ATAC) trial, are not supportive of this CYP2D6 genotyping in the adjuvant setting [42,43]. They found that there is no relationship between the re-occurrence of breast cancer and lower endoxifen concentration due to different categories of CYP2D6 phenotype. The outcomes of these trials drew criticism from several researchers. The outcomes were influenced by all these elements. First, patients were not receiving tamoxifen monotherapy; they were concurrently receiving additional systematic therapy or were switched to aromatase inhibitors in the adjuvant setting; second, the follow-up period was brief, being only three years; and third, the study was not powered to investigate the association between clinical outcome and the endoxifen concentration [42].

CYP2D6 phenotype is usually determined by genotyping based on single-nucleotide polymorphism analysis and using the probe drugs which are CYP2D6 substrates. This CYP2D6 substrate-based phenotyping is quantified in terms of metabolic ratio, expressed as concentration of unchanged probe drug divided by the concentration of metabolite at any specified time after the drug administration [40]. This method of phenotyping using probe is used widely than the determination of phenotypes from SNP-based genotypes.

The IIV on apparent clearance of tamoxifen and endoxifen formation ranged from 25% to 39.9% and 46% to 56.2%, respectively. In Puzkiel et al. and Dahmane et al.’s studies, IIV ranges for K_TAM/NDT_, K_TAM/4′-OHTAM_, and K_NDT/Z’-ENDO_ varied from 16% to 31.6%, 19.6% to 26%, and 47.4% to 59%, respectively. CYP2D6 phenotype was able to explain 39% of the IIV on endoxifen formation, leaving a large proportion of variability in endoxifen plasma concentrations [14]. As body weight increased, tamoxifen clearance also increased; therefore, body weight accounted for 19% of the variability in tamoxifen clearance [32].

Other than CYP2D6 phenotype, various other factors also influenced the endoxifen concentrations, some of which include CYP3A4, SULT1A1 enzyme activity, age, menopausal status, ethnicity, and concomitant drugs, especially CYP2D6 inhibitors or inducers [43,44]. CYP3A4/5 enzymes are also responsible for the conversion of tamoxifen to its active metabolites. It has been reported that CYP3A4*22 polymorphism has reduced metabolic activity, and CYP3A5*6 and CYP3A5*3 polymorphisms result in no metabolic activity. This leads to the lower concentration of endoxifen and N-desmethyl tamoxifen, leading to poor response [8].

A few studies have reported that UDP glucuronosyltransferases (UGTs), sulfotransferases (SULTs), and demethylases are the most influential enzymes in the elimination and inactivation of endoxifen. In particular, the SUL1A1 enzyme, a subgroup of SULTs enzyme, has a vital role in the deactivation of endoxifen into endoxifen sulfate. The genetic polymorphism in this enzyme is associated with the endoxifen concentration and the survival outcomes. A study by Nowell et al. identified poor overall treatment outcomes in patients with UGT2B15*2, SULT1A1*1/*1 or SULT1A1*1/*2 and SULT1A1*2/*2 carriers [45,46].

Age has been found to be one of the critical factors in both tamoxifen and endoxifen metabolism. Three studies had reported that an increase in age resulted in a decrease in the tamoxifen apparent clearance [32,33,34]. Puszkiel et al. identified that the elimination of tamoxifen is twofold higher in patients who are thirty years of age, whereas it is 25% lower in patients who are ninety years of age. Age explained 21% of variability in the tamoxifen clearance in a study by Schoell et al. Some studies reported that tamoxifen’s active metabolites concentration is higher in the older population (>69 years) than in the younger population [47,48,49]. However, a study by Antunes et al. reported an inverse correlation to the above statement [50]. The difference in these results could be attributed to human physiological changes, such as aging, reduced metabolic activity or enzyme activity, co-morbidities, polypharmacy, or menopause status [16].

According to research, there is an inverse relationship between body mass index (BMI) and tamoxifen and its metabolites’ concentration in plasma. This proportionality was explained by the relationship of higher Vd in people with a greater BMI, resulting in lower drug concentration. Despite this correlation, there are no guidelines for tamoxifen dose change based on BMI in clinical settings [17,51,52].

Clinical depression is believed to affect 10–25% of female breast cancer patients. To treat the condition, individuals are given selective serotonin norepinephrine reuptake inhibitors (SNRIs) and/or selective serotonin reuptake inhibitors (SSRIs). Nonetheless, the co-administration of these CYP2D6 inhibitors has resulted in a significant drug–drug interaction (DDI). The result was a significant drop in endoxifen concentration, leading to sub-therapeutic concentrations [53,54]. Schulze et al. reported that the SSRI co-administration and rifampicin (CYP2D6 inducer) in NMs resulted in 65.4% and 2.42% reductions in endoxifen formation [33]. There is a considerable variation in the plasma concentration of tamoxifen and its metabolites in pre-menopausal and post-menopausal patients. It has been reported that the steady-state concentration of tamoxifen is increased by 70–80% and endoxifen by 135% in post-menopausal women diagnosed with breast cancer compared to pre-menopausal women. This variation is believed to be because of the difference in the hormonal status of these groups [55].

A few studies suggested that compared to traditional dosing, simulation-based drug dosing for the patient based on their characteristics would increase efficacy and reduce toxicity [32,33,34]. Pharmacometrics simulation aids in the interpretation of models. Another essential role of simulation is to predict drug effects under various unobserved conditions. When making a simulation-based decision, it is preferable to consider interindividual variation—a real biological phenomenon—rather than the more commonly used deterministic simulation, which is based solely on fixed effect parameters and ignores random effect [56].

## 6. Limitations

Most of the studies utilized clinical data from the other studies in developing the PopPK model. So, there could be bias in the data used for modeling, which cannot be identified in the selected articles. The review included only studies in English and excluded other languages. The review only summarizes the various PopPK models but does not explain the model’s generalizability, which can be used for informed dosing decisions in clinical practice for these populations.

## 7. Conclusions

This review states that the CYP2D6 phenotype influences tamoxifen and its metabolites’ pharmacokinetics. However, anthropometric characteristics and the polymorphism of additional genes that encode the enzymes for tamoxifen metabolism all have an impact on these parameters. Most of the studies reported a two-compartment structural model to explain the pharmacokinetics parameters of tamoxifen and its metabolite in breast cancer patients. External validation of these models will help identify the generalizable or integrated model that can be utilized for model-based informed dosing of tamoxifen in the breast cancer population.

## Figures and Tables

**Figure 1 biology-12-00051-f001:**
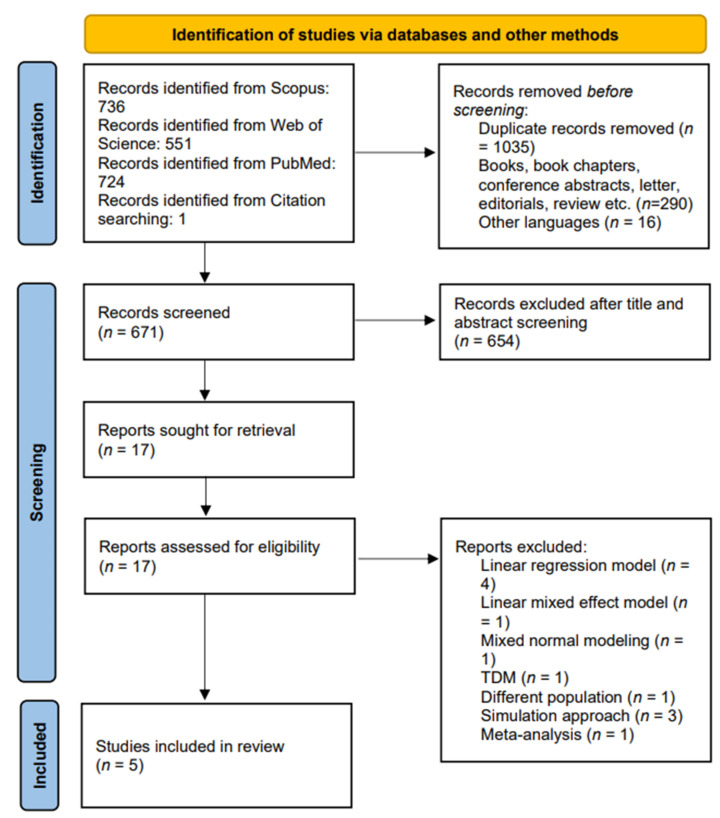
PRISMA flowchart.

**Table 1 biology-12-00051-t001:** Checklist for assessing the quality of tamoxifen PopPK studies.

Quality Criteria	Ter Heine et al. 2014 [15]	Schoell et al. 2020 [32]	Schulze et al. 2020 [33]	Puszkiel et al. 2021 [34]	Dahmane et al. [35]	Compliance Rate of Each Criterion (%)
**Title**						
The title identifies the drug(s) and patient population(s) studied	×	√	×	√	√	60
**Abstract**						
Name of the drug(s) studied	√	√	√	√	√	100
Patient population studied	√	√	√	√	×	80
Primary objective(s)	√	√	√	√	√	100
Major findings	√	√	√	√	√	100
**Background/introduction**						
Study rationale	√	√	√	√	√	100
Specific objectives/hypothesis	√	√	√	√	√	100
**Methods**						
Ethics approval	√	√	√	√	√	100
Eligibility criteria of study participants	√	√	√		√	80
Co-administration or food	×	×	×	√	×	20
Dosing/frequency/formulation	×	√	√	√	√	80
Sampling time and frequency	√	×	√	×	√	60
Type of sample	√	√	√	√	√	100
Bioanalytical method	√	√	√	√	√	100
Statistical method and software used	×	×	×	√	×	20
Modeling software	√	×	×	√	√	60
Modeling assumptions made	×	×	√	√	√	60
Estimation method(s) used	√	×	√	×	√	60
Structural model	√	√	√	√	√	100
Covariates tested	√	√	√	√	√	100
Covariate analysis strategy	√	√	√	√	√	100
Residual error model	√	√	√	√	√	100
Methods for final model evaluation	√	√	√	×	√	80
External model validation	NA	√	NA	NA	NA	100
Model selection criteria (OFV/AIC, etc.)	√	√	√	√	√	100
Number of study subjects	×	√	√	√	√	80
Number of samples used for analyses	×	√	√	×	√	60
Equations for all model structures and covariate relationships	√	×	√	×	×	40
**Results**						
Demographics details and clinical variables	√	√	√	√	√	100
Concentration vs. time plot	×	×	×	×	×	0
Schematic of the final model	√	√	√	√	√	100
Table of final model parameters	√	√	√	√	√	100
Summary of the model building process and the derived final model	√	√	√	√	√	100
Final model evaluation plots	√	√	√	√	√	100
A description of simulation results or scenarios (if applicable)	NA	√	√	√	×	75
**Discussion/conclusion**						
Study limitations	√	×	×	√	×	40
Study findings	√	√	√	√	√	100
**Total compliance rate of each study (%)**	77.1	75.6	83.3	80.5	80.5	

√ denotes study reported the quality criteria, × denotes study did not report the quality criteria, and NA Not applicable.

**Table 2 biology-12-00051-t002:** Demographics of selected studies.

Variables	Ter Heine et al. 2014 [15]	Schoell et al. 2020 [32]	Schulze et al. 2020 [33]	Puszkiel et al. 2021 [34]	Dahmane et al. [35]
No. of subjects	40	452	468	928	97
No. of samples per patient	9	1–9	1–27	7	5
Total no. of samples	680 (349 + 331)	NA	3554	27,433	457
Bio-analytical method	UPLC-MS/MS	HPLC-MS/MS,	HPLC-MS/MS,	UPLC-MS/MS	HPLC-MS/MS
		UPLC-MS/MS	UPLC-MS/MS		
Type of sample	Plasma	Serum	Serum	Plasma	Plasma
		Plasma	Plasma		
Age (years)	53 (22–71)	64 (25–95)	64 (25–95)	48 (25–84)	50 (32–78)
Height (m)	1.69 (1.56–1.79)	NA	NA	NA	1.65 (1.51–1.83)
Weight (kg)	72.7 (48.5–114)	70 (42–150)	NA	64 (40–131)	65 (47–116)
Tamoxifen dose (%)					
20 mg QD	70	98.9	96	100	100
40 mg QD	30	1.1	4	NA	
CYP2D6 phenotype (%)					
Ultrarapid metabolizer (UM)	2.5	NA	1	3.7	3
Normal metabolizer (NM)	50	53.5 (including UM)	78	83.3	62
Intermediate metabolizer (IM)	45	34.5	8	8.6	31
Poor metabolizer (PM)	2.5	5.53	6	4.4	4
Missing	NA	6.42	7	NA	NA
Menopause status (%)	NA	NA	NA	NA	
Pre-menopause					51.5
Post-menopause					48.4
Missing					

NA Not Applicable.

**Table 3 biology-12-00051-t003:** Parameters of selected studies.

Study and Year	Model Structure	External Validation	Residual Variability	Parameter Estimates	Significant Covariates
**Ter Heine et al. 2014** [15]	Two-compartment model with first-order absorption and elimination	No	Proportional error	K_a_ (1/h)—1.90	CYP2D6 phenotype
				T_lag_ (h)—0.455	CYP3A4
				Q_1_ (l/h)—61.8	
				V_d_ tamoxifen (l)—753	
				CL_TAM_ (l/h)—9.34	
				CL_MET_ (l/h)—0.324	
				Clendo(l/hr) = 5.1	
				Vdendo(l) = 400	
				Theta2D6,1 = 0.262	
				Theta3A4,1 = 0.157	
**Schoell et al. 2020** [32]	Two-compartment model with first-order absorption and elimination	Yes	Proportional error	K_a_ (1/h)—1.08 (Fixed)	Age
				T_lag_ (h)—0.442 (Fixed)	Body weight
				V_TAM_/F (l)—912 (Fixed)	CYP2D6 phenotype Activity score
				CL30/F (l/h)—5.10 (Fixed)	
				V_ENDX_/F (l)—400 (Fixed)	
				CL20/F (l/h)—5.07	
				CL23/F (l/h)—0.459	
				CL20/F_Age: −0.17	
				CL20/F_Bodyweight: 0.284	
				CL23/F_AS: 0: −0.759	
				CL23/F_AS: 0.5: −0.598	
				CL23/F_AS: 1: −0.347	
				CL23/F_AS: 1.5: −0.16	
				CL23/F_AS: 2.5–3: 0.302	
**Schulze et al. 2020** [33]	Two-compartment model with first-order absorption and elimination	No	Proportional error	K_a_ (1/h)—1.78	Age
				T_lag_ (h)—0.389	CYP2D6 phenotype
				V_TAM_/F (l)—1120	Co-medication (Rifampicin/SSRI)
				CL30/F (l/h)—5.10 (Fixed)	
				V_ENDX_/F (l)—400 (Fixed)	
				CL20/F (l/h)—5.77	
				CL23/F (l/h)—0.493	
				Vtam/F_Rif: 0.581	
				CL20/F_Rif: 6.51	
				CL20/F_Age: −0.886	
				CL23/F_AS: 0: −0.722	
				CL23/F_AS: 0.5: −0.510	
				CL23/F_AS: 1: −0.323	
				CL23/F_AS: 1.5: −0.211	
				CL23/F_AS: 2.5–3: 0.533	
				CL23_SSRI: −0.654	
				CL23_Rif: 1.18	
**Puszkiel et al. 2021** [34]	Seven-compartment model with first-order absorption and elimination	No	Proportional error	K_a_ (1/h)—0.90 (Fixed)	CYP3A4*22 genotype
				V_TAM_ (l)—1380	Age
				K_TAM/NDT_ (1/h)—5.20 × 10^−3^	CYP2D6 phenotype
				Effect of CYP3A4*22 genotype: 0.773	CYP2C19*2 genotype
				Effect of age: −0.298	CYP2B6*6/*6 genotype,
				K_TAM/4-OHTAM_ (1/h)—3.72 × 10^−5^	Co-medication (CYP2D6 inhibitors)
				Effect of CYP2D6 IM or PM phenotype: 0.768	Body weight
				Effect of CYP2D6 missing phenotype: 1.25	
				Effect of CYP2C19*2 genotype: 0.866	
				Effect of age: −0.547	
				K_TAM/4’-OHTAM_ (1/h)—6.16 × 10^−8^	
				K_TAM/NOX-TAM_ (1/h)—2.48 × 10^−7^	
				Effect of CYP2B6*6/*6 genotype: 0.766	
				Effect of age: −0.296	
				KNDT/ENDO:	
				CYP2D6 UM (h^−1^): 6.87 × 10^−4^	
				CYP2D6 NM (h^−1^): 5.42 × 10^−4^	
				CYP2D6 IM (h^−1^): 2.86 × 10^−4^	
				CYP2D6 PM (h^−1^): 0.88 × 10^−4^	
				Missing CYP2D6 phenotype(h^−1^): 6.04 × 10^−4^	
				Effect of weak/moderate CYP2D6 inhibitor in NM and UM: 0.680	
				Effect of potent CYP2D6 inhibitor in NM and UM: 0.434	
				Effect of age: −0.480	
				K_NDT/Z’-ENDO_ (1/h)—4.08 × 10^−7^	
				K_4-OHTAM/ENDO_ (1/h)—1.81 × 10^−3^	
				K_e,NDT_ (1/h)—2.46 × 10^−3^	
				Effect of CYP3A4*22 genotype: 0.812	
				Effect of body weight: 0.245	
				K_e,ENDO_ (1/h)—7.93 × 10^−3^	
				K_e,4′-OHTAM_ (1/h)—2.01 × 10^−6^ (Fixed)	
				K_e,NOX-TAM_ (1/h)—1.77 × 10^−6^ (Fixed)	
				K_e,Z’-ENDO_ (1/h)—1.08 × 10^−5^ (Fixed)	
**Dahmane et al.** [35]	Four-compartment model with first-order absorption and elimination	No	Proportional error	CL_TAM_/F (l/h)—5.8	Age
				θAge: 0.5	Metabolic ratio
				θMR: 0.16	Compliance
				θCompliance: 0.09	CYP2D6 phenotype
				V2/F (l)—724	Co-medication (CYP2D6 inhibitor)
				K_a_ (1/h)—0.7 (Fixed)	
				K_23_ (1/h)—7.07 × 10^−3^	
				θMR: 0.07	
				K_24_ (1/h)—5.49 × 10^−5^	
				θCYP2D6 PM/IM: 0.26	
				K_35_ (1/h)—2.84 × 10^−4^	
				θCYP2D6 PM: 0.96	
				θCYP2D6 IM: 0.56	
				θpotent 2D6 inhibitor: 0.85	
				θmoderate 2D6 inhibitor: 0.41	
				K_45_ (1/h)—0.015	
				CL_NDT_/F (l/h)—3.4	
				CL_4-OHTAM_/F (l/h)—2.9	
				CL_END_/F (l/h)—6.2	

K_a_—absorption rate constant, T_lag_—lag time, CL/F—apparent clearance, V/F—apparent volume of distribution, Q—intercompartmental clearance, CL—clearance, Vd—volume of distribution, Ke—elimination rate constant.

## Data Availability

Not applicable.

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
