# Peer review of "Factors Influencing Pharmacokinetics of Tamoxifen in Breast Cancer Patients: A Systematic Review of Population Pharmacokinetic Models"

_biology, 2022, doi:10.3390/biology12010051_

Round 1

Reviewer 1 Report

Comment 1: Supplementary file:  The search equation includes NONMEM, in the first item of the search. I understand that NONMEM is used for most of the pharmacometrician, and it is an indubitable sign of PopPk modelling, but why don’t you include other widely used software, as Monolix?

Comment 2: in line 288 the author stated that “Most studies (n = 3) used NONMEM for PopPK modeling.”. Which studies? And what software do the other 2 studies used?

Comment 3: According to table 1, there are 3 studies that included simulations. It would be interesting to describe and discuss these simulations in the review.

Comment 4: In line 329, please define what kind of external validation was performed.

Comment 5: In the Discussion section (line 335), the authors mention the absence of correlation between CYP2D6 phenotype and outcomes in prospective clinical trials. However, in the next paragraphs it is said that the CYP2D6 phenotype is clearly related to the probability of having sub-target endoxifen steady-state concentrations. What is the opinion of the authors about the lack of significance in the clinical trials?

Comment 6: The pharmacokinetic models found in literature are not described enough, please include a figure or table where the structural model, including covariates, are showed.

Author Response

Dear Reviewer,

Thank you for your insightful comments regarding this systematic review. Please find the responses to each comment in the attachment.

Reviewer 2 Report

Thanks for this manuscript but the rational is not well explain in introduction part and following are the few comments:

What is the value addition of this study in literature?

Why did authors exclude conference abstracts and short communications?

Why did you attach the data extraction file?

What is the reason not to record the race, ethnicity, and geography location?

Author Response

(The authors gave the same response as above.)

Reviewer 3 Report

The authors carry out a good systematic review of the different PopPKs published on tamoxifen and its active metabolites, developed in breast cancer patients. The work is very interesting because it makes a critical and very useful analysis of the covariates that significantly affect the PK behavior of this drug.

Comments and Suggestions for Authors:

The sample size of the population ranged between 40 to 928 (table 2). This is a very important difference that should lead to the exclusion of the PopPK model developed with a sample size as low as 40.

One limitation of the work is that they only evaluate models based on non-linear mixed effects modeling with a parametric approach, and it would have been advisable to also review others that use the non-parametric approach, although these are much less commonly used.

In the discussion, the range of % contribution to IIV of the CL of tamoxifen of each of the covariates found in the different PopPK models  evaluated should be commented in a little more detail.

It would be highly advisable to elaborate a table summary with the range of the contribution of different covariates with influence on the PK behavior of tamoxifen in order to establish “a priori” dosage guidelines which would be very useful to apply in the clinical context.

Lines 93-94: “These metabolites….activity”. the authors should clarify that the affinity of these metabolites is referred to tamoxifen.

Line 385: “drop in drug concentration” Specify the pharmacologically active entity to which “drug” refers

Lines 196-97: Why do authors exclude works that perform meta-analysis when these studies provide very good information?

Table 2. This table should show the range of the number of samples per patient, as indicated on line 229 of the text.

Lines 298-299: You must specify the type of distribution volume of the two-compartment model. It's from the central compartment, is it the Vdss?

Lines 316-317: The age range in which the tamoxifen clearance decrease by 9% for increase in every ten years of age should be specified.

Lines 324-326: The clinical significance of these low IIV values (4, 8 and 1%) explaining the variability of tamoxifen apparent clearance should be discussed.

Line 329: Results of this external validation should be commented.

Lines 344-345: In my opinion this phrase should emphasize that the CYP2D6 genotype is the most important covariate since it is capable of explaining most of the IIV in tamoxifen clearance (39%). This sentence could be rewritten as follows: Only CYP2D6 genotype was able to explain 39% of the IIV….

Author Response

(The authors gave the same response as above.)

Reviewer 4 Report

The manuscript describes the results of search for articles concerning population pharmacokinetics of tamoxifen in breast cancer patients.  The aim of the study is not entirely clear. The variables affecting pharmacokinetics of tamoxifen are well known (and presented in the Introduction of the maniscript)  and many articles describing this issue are available. In addition, the Authors in the Discussion section  cited these articles despite the fact that NONMEM was not used by their authors to identify factors leading to variability in pharmacokinetics of tamoxifen.

Other concerns

The study selection criteria included non-linear mixed effects modeling approach and the authors included only these papers where NONMEM was used as modelling software. Other software, e.g. Monolix also use this approach.   

The discussion is very short and seems irrelevant to the aim of the study. The Authors present factors influencing tamoxifen PK citing papers other than selected based on the selection criteria. In addition these papers did not employ population modeling to identify these factors.

To make these manuscript more interesting and useful for a wider group of readers all existing population models should be presented and discussed indicating their strength and weaknesses as well as their usefulness for dose optimization in various populations.

Author Response

(The authors gave the same response as above.)

Round 2

Reviewer 1 Report

Thanks for the response. The comment 6 has not been answered properly, however, I would consider accept in the present form.

Reviewer 2 Report

Thanks for the updated manuscript. 

Reviewer 4 Report

The Authors have improved the manuscript, especially the Discussion section and adequately addressed most of my concerns.